# Large-Effect QTLs for Titratable Acidity and Soluble Solids Content Validated in 'Honeycrisp'-Derived Apple Germplasm

**Baylee A. Miller, Sarah A. Kostick \* and James J. Luby**

Department of Horticultural Science, University of Minnesota, Saint Paul, MN 55108, USA;
mill6461@umn.edu (B.A.M.); lubyx001@umn.edu (J.J.L.)
* Correspondence: kosti028@umn.edu

**Abstract:** Fruit acidity and sweetness are important fruit quality traits in the apple and are therefore targets in apple breeding programs. Multiple quantitative trait loci (QTLs) associated with titratable acidity (TA) and soluble solids content (SSC) have been previously detected. In this study a pedigree-based QTL analysis approach was used to validate QTLs associated with TA and SSC in a 'Honeycrisp'-derived germplasm set. TA and SSC data collected from 2014 to 2018 and curated genome-wide single nucleotide polymorphism (SNP) data were leveraged to validate three TA QTLs on linkage groups (LGs) 1, 8, and 16 and three SSC QTLs on LGs 1, 13, and 16. TA and SSC QTL haplotypes were characterized in six University of Minnesota apple breeding families representing eight breeding parents including 'Honeycrisp' and 'Minneiska'. Six high-TA haplotypes, four low-TA haplotypes, 14 high-SSC haplotypes, and eight low-SSC haplotypes were characterized. The results of this study will enable more informed selection in apple breeding programs.

**Keywords:** *Malus domestica* Borkh.; quantitative trait locus; *Ma*; *Ma3*; pedigree-based QTL analysis

## 1. Introduction

The balance of acidity and sweetness is an important part of the apple (*Malus domestica* Borkh.) eating experience and product acceptance [1] and is therefore an important breeding target. In recent decades, apple breeding programs have implemented the use of DNA information to inform breeding decisions (i.e., DNA-informed breeding). Due to the long juvenility phase of the apple, acidity and sweetness traits are ideal targets for DNA-informed breeding (e.g., marker-assisted parent and/or seedling selection).

Standardized phenotyping is required for quantitative trait locus (QTL) mapping but is a challenge for traits that can only be phenotyped in mature trees that are producing fruit. The long juvenility phase of apple trees, the strong influence of the environment, and the often-limited crop load of seedling trees make it challenging to evaluate acidity and sweetness. Common methods for phenotyping acidity include sensory evaluation [2], measuring titratable acidity (TA) [3] and pH [4], and common methods for phenotyping sweetness include measuring individual sugars and soluble solids content (SSC) [5].

Multiple QTLs associated with fruit acidity components have been previously reported [2–4,6–12]. QTL mapping studies have confirmed a major QTL for acidity on linkage group (LG) 16 (*Ma* locus) [2–4,6,7,11] as well as other QTLs on LG8 (*Ma3* locus) [2,3,6,8,11] and on LG1 (*Ma5* locus) [2,9,10]. Several other QTLs for acidity have been identified on LG4 [12] and LG6 [2,12]. Verma et al. [3] characterized the haplotype effects at the *Ma* and *Ma3* loci in a limited set of germplasm that can be utilized by breeders to predict cross outcomes more accurately.

QTL mapping studies for sweetness-related components including fructose, glucose, sucrose, sorbitol, and SSC have reported QTLs on LG1 [5,8,10], LG13 [5], and LG16 [5,13]. QTLs for sweetness-related components have been identified on nearly every linkage group of the apple [5–8,10,13], which demonstrates the highly quantitative nature of sweetness components.

The goal of this study was to improve the understanding of the genetic factors that influence TA and SSC in a 'Honeycrisp'-derived germplasm set representative of the UMN apple breeding program. High-quality, genome-wide single nucleotide polymorphism (SNP) data [14,15] and FlexQTL™ software [16,17] were leveraged in this study to (1) detect QTLs associated with TA and SSC and (2) characterize QTL haplotype effects. We hypothesized that the previously identified QTLs for TA and SSC would be validated in the UMN germplasm set.

## 2. Materials and Methods

### 2.1. Plant Material

The germplasm in this study consisted of 974 individuals (i.e., unselected offspring, cultivars, and advanced selections) from UMN breeding germplasm planted at the University of Minnesota (UMN) Horticulture Research Center in Chanhassen, Minnesota (44.87° N, −93.63° E). This study focused on six pedigree-connected full-sib families (*n* = 47 to 133) that represented eight breeding parents ('Honeycrisp', 'Minneiska', 'Minnewashta', MN1836, MN1915, MN1965, 'MN55', and 'Wildung'). Offspring were represented by one or two replicate trees propagated on B.9 rootstock (Table S1). Fruit from all offspring were harvested in at least one year from 2014 to 2018.

### 2.2. Phenotypic Data

Individuals were evaluated using the RosBREED standardized apple phenotyping protocol (described by Evans et al. [18]). A minimum of three fruit were harvested per tree when the average starch pattern index rating [19] ranged from four to six on an eight-point scale. Fruit from biological replicate trees (i.e., 'Honeycrisp' × 'Minnewashta' offspring) were harvested separately. Fruit collected from individual trees were juiced together within four days of harvest. Juice samples were frozen and stored at −80 °C until TA and SSC could be measured.

Juice samples were thawed to room temperature (21 °C) and homogenized prior to measuring TA and SSC. An automatic titrator (Mettler Toledo T50 and InMotion Flex Autosampler) was used to measure TA by using 5 mL of juice titrated with 0.1 M NaOH until the pH reached 8.2. A handheld refractometer (Atago Digital Hand-held PAL-1, https://www.atago.net/en/products-index.php?key=ABS42888; accessed on 18 July 2022) was calibrated using deionized water and then approximately 0.2 mL of juice were placed onto the device to measure SSC. Two technical replicates per juice sample were evaluated. Within a given year, measurements from biological replicates were averaged to record a single measurement per individual. Across-year corrected trait means were estimated for each individual by calculating year effects for the years 2014 to 2018 and correcting an individual's overall mean. Corrected means were used as phenotypic values in the QTL analyses similar to the use of the best linear unbiased predictions as phenotypic values in previous studies [20,21].

### 2.3. Genotypic Data

All individuals were genotyped via either the International RosBREED SNP Consortium 8K Illumina Infinium® array v1 [14] or the Illumina Infinium® 20K array [15]. SNP markers (*n* = 2113) common to both arrays were utilized in the QTL and haplotype analyses. SNP marker calling, filtering, and curation were performed using the methods described by Vanderzande et al. [22]. The genetic map used in this study was previously described by Howard et al. [23].

### 2.4. QTL Mapping and Haplotype Analyses

#### 2.4.1. QTL Analyses

FlexQTL™ software [16,17] was utilized to perform the QTL analyses for TA and SSC. Separate QTL analyses were performed for each trait within each year (2014–2018) and with adjusted across-year means. Convergence was considered reached when the effective chain



sizes for each parameter setting reached at least 100, which was achieved with Markov chain lengths of 150,000. An additive model was used. The results were evaluated by identifying LGs with a Bayes factor parameter (BF; $2lnBF_{10}$) greater than 5 (strong), or greater than 10 (decisive) [24]. Each QTL region on each LG was determined using the postgenome.csv file to identify stretches of 2 cM bins with a BF > 5 [20,25]. The bin with the highest BF of each QTL region was determined to be the most likely QTL position (i.e., QTL peak).

### 2.4.2. Haplotype Analyses

QTL regions targeted for haplotype analysis had BF > 5 and were consistent across at least two different years or were detected in the across-year dataset. Successive SNP markers were selected for haplotyping based on proximity to QTL peaks. Marker phasing was performed using FlexQTL™ software. For TA QTL regions that colocalized with previously reported TA QTLs, the QTL regions were selected for haplotyping to be consistent with previously haplotyped regions. Within the six families, uninformative SNPs and SNPs with redundant segregation patterns were excluded. Offspring with recombinant QTL haplotypes were removed from further analyses due to low relative haplotype representation. Parental haplotypes were traced through extended pedigrees, reconstructed by Luby et al. [26], to the furthest known ancestor. Haplotypes that were traced to a common ancestor were considered identical by descent, whereas haplotypes that could not be traced to a known common ancestor were defined as identical by state.

Analyses of variances across and within families were used to determine if haplotypes at a given QTL had significant effects on TA or SSC. Mean separation was performed using Tukey's honest significant difference (HSD) test. Relative haplotype effects (i.e., high, low, or neutral relative TA or SSC) were assigned to haplotypes for each trait using the mean separation groups. To examine the effects of diplotypes at a given locus, individuals were grouped by their functional diplotypes. Trait means of individuals homozygous for high- or low-effect QTL haplotypes were compared to trait means of individuals homozygous for neutral-effect haplotypes or heterozygous for functional haplotypes.

## 3. Results

### 3.1. Phenotypic Data

Quantitative variation for TA and SSC was observed within and among full-sib families. The across-year adjusted TA means ranged from 1.63 to 14.80 g/L. The across-year adjusted SSC means ranged from 7.60 to 18.90 °Brix.

### 3.2. QTL Detection

The QTLs for TA were detected on LGs 1, 7, 8, 10, and 16 with strong (BF > 5) to decisive (BF > 10) evidence using the across-year adjusted dataset and/or in at least one year (Table 1). The QTLs on LG1, LG8, and LG16 were consistent across multiple years.

The QTLs for SSC were detected on LGs 1, 9, 12, 13, 15, 16, and 17 with strong (BF > 5) to decisive (BF > 10) evidence using the across-year adjusted dataset and/or in at least one year (Table 1). The QTLs on LG1 and LG13 were consistent across multiple years.

### 3.3. Haplotype Analyses for Titratable Acidity QTLs

### 3.3.1. LG1 Titratable Acidity QTL

Six LG1 QTL haplotypes, constructed with six SNP markers spanning a 5.3 cM distance (1.42 Mbp), segregated in the target families for TA and varied significantly in terms of their TA levels ($p < 0.0001$). One haplotype (1A) was associated with a high relative TA ($\mu = 6.9$ g/L), two haplotypes (1E, 1F) were associated with a low relative TA ($\mu = 6.0$ g/L and $\mu = 5.8$ g/L, respectively), and three haplotypes (1B, 1C, 1D) were associated with a moderate relative TA ($\mu = 7.2$ g/L, $\mu = 6.5$ g/L, $\mu = 6.3$ g/L, respectively) (Table 2).

**Table 1.** Summary of TA and SSC QTL analysis results across years and datasets. QTLs with strong (BF > 5) to decisive (BF > 10) evidence are reported.

| Trait [z] | Year [y] | LG [x] | BF [w] | QTL Region (cM) [v] | QTL Mode (cM) [u] | Position (Mbp) [t] |
|---|---|---|---|---|---|---|
| TA | 2017 | 1 | 6.6 | 57–63 | 62 | 31.1–32.5 |
| | 2015 | | 11.4 | 31–41 | 39 | 23.6–24.2 |
| | 2018 | | 31.1 | 49–53 | 51 | 28.4–30.0 |
| | 2014–2018 | | 31.1 | 47–53 | 47 | 27.7–30.0 |
| | 2014 | | 31.3 | 45–55 | 53 | 26.9–30.9 |
| | 2014 | 8 | 8.2 | 3–9 | 5 | 1.3–2.7 |
| | 2014–2018 | | 30 | 23–33 | 29 | 7.4–12.0 |
| | 2018 | | 30.8 | 27–33 | 29 | 9.4–12.0 |
| | 2017 | | 31 | 31–35 | 33 | 11.0–12.6 |
| | 2014–2018 | 10 | 7.9 | 19–25 | 23 | 9.4–18.9 |
| | 2017 | 16 | 9.2 | 3–13 | 13 | 2.0–4.2 |
| | 2014 | | 10.6 | 7–11 | 9 | 3.0–3.6 |
| | 2015 | | 29.9 | 11–13 | 11 | 3.6–4.2 |
| | 2018 | | 30.8 | 7–11 | 9 | 3.0–3.6 |
| | 2014–2018 | | 30.8 | 9–11 | 9 | 3.2–3.6 |
| SSC | 2014–2018 | 1 | 7.6 | 47–53 | 51 | 27.7–30.0 |
| | 2014 | 9 | 5.2 | 23–25 | 25 | 8.0–8.5 |
| | 2017 | 12 | 5.5 | 25–27 | 27 | 12.9–17.2 |
| | 2016 | 13 | 7 | 43–47 | 43 | 12.2–14.1 |
| | 2018 | | 15.1 | 65–69 | 67 | 23.4–26.1 |
| | 2014–2018 | | 15.1 | 63–67 | 65 | 22.1–25.7 |
| | 2014–2018 | 15 | 7.5 | 35–43 | 37 | 9.8–12.2 |
| | 2017 | 16 | 7.1 | 3–9 | 7 | 2.0–3.2 |
| | 2014–2018 | | 6.9 | 1–9 | 7 | 0.9–3.2 |
| | 2017 | 17 | 5.8 | 33–39 | 35 | 11.8–13.9 |

[z] Titratable acidity (TA) and soluble solids content (SSC); [y] year or across-year corrected mean; [x] linkage Group; [w] linkage-group-wise Bayes factor (BF; $2\ln BF_{10}$) value for 1 QTL vs. 0 QTL model, with BF > 2, 5, and 10 indicating positive, strong, or decisive evidence for the presence of one QTL, respectively; [v] QTL region in centimorgans (cM), 2 cM bins with BF >5; [u] QTL mode, most likely QTL location defined as the 2 cM bin with the highest BF; [t] approximate physical position of QTL interval, estimated from physical positions on GDDH13v1.1 reference genome [27].

**Table 2.** Summary of QTL haplotype effects for the LG1, LG8, and LG16 TA QTLs. Across-year adjusted TA means for six families were used.

| LG | Haplotype Name | Haplotype Sequence [z] | Mean (g/L) | Variance | No. Offspring | Relative Effect | Reported Effect [y] |
|---|---|---|---|---|---|---|---|
| 1 | 1A | ABBABA | 6.9 | 5.0 | 255 | High | - [x] |
| | 1B | ABBAAA | 7.2 | 5.6 | 39 | Neutral | - |
| | 1C | BABABB | 6.5 | 4.8 | 152 | Neutral | - |
| | 1D | BBAABA | 6.3 | 3.3 | 214 | Neutral | - |
| | 1E | BBABAB | 6.0 | 4.6 | 154 | Low | - |
| | 1F | BBABAA | 5.8 | 5.8 | 33 | Low | - |

**Table 2.** *Cont.*

| LG | Haplotype Name | Haplotype Sequence [z] | Mean (g/L) | Variance | No. Offspring | Relative Effect | Reported Effect [y] |
|---|---|---|---|---|---|---|---|
| | 8A | BBABBBBAABAAAB | 7.4 | 5.7 | 52 | High | Unknown [w] |
| | 8B | AAAABAAABAAAAA | 7.1 | 6.5 | 51 | High | Unknown |
| | 8C | BABBBBBBABAABB | 7.0 | 4.5 | 212 | High | *q* [v] |
| | 8D | BABAABBBABABAB | 6.9 | 4.4 | 20 | Neutral | Unknown |
| 8 | 8E | BAABBAAABAAAAA | 6.6 | 2.3 | 73 | Neutral | Unknown |
| | 8F | AABBABBBABABAA | 6.4 | 4.4 | 161 | Neutral | *q* |
| | 8G | BAABBBBAABABAB | 6.3 | 6.5 | 41 | Neutral | Unknown |
| | 8H | BAABBBAAAAAAAB | 6.4 | 8.2 | 24 | Neutral | *q* |
| | 8I | BAABBBAAABBABA | 5.7 | 4.8 | 184 | Low | *q* |
| | 16A | BABAAB | 6.8 | 5.0 | 205 | High | *Q* [u] |
| | 16B | ABBBAA | 6.8 | 5.4 | 207 | High | *Q* |
| | 16C | BBBBBB | 6.6 | 3.8 | 143 | Neutral | *Q* |
| 16 | 16D | BBAAAB | 6.5 | 4.6 | 141 | Neutral | *Q* |
| | 16E | ABBBBA | 6.0 | 2.7 | 43 | Neutral | *q* |
| | 16F | BBBBAA | 6.0 | 4.4 | 104 | Neutral | Unknown |
| | 16G | BBAAAB | 5.9 | 4.8 | 23 | Neutral | *Q* |
| | 16H | ABBBAA | 4.8 | 4.1 | 29 | Low | *q* |

[z] Information on SNP markers used for haplotyping at the LG1, LG8, and LG16 TA QTLs is presented in Table S2. [y] Reported effects are in reference to the results from Verma et al. [3]; [x] no characterization for the LG1 locus haplotypes was previously reported; [w] unknown indicates that the haplotype was not previously reported; [v] *q* is the low-relative-effect allele reported by Verma et al. [3]; [u] *Q* is the high-relative-effect allele reported by Verma et al. [3].

The high-TA haplotype 1A of the breeding parents 'Honeycrisp', 'Minneiska', and 'MN55' was traced to a recombination event between the two haplotypes of 'Grimes Golden'. The low-TA haplotype 1F of the breeding parent MN1965 was traced to 'Aspa'. The low-TA haplotype 1E of the breeding parents 'Honeycrisp', MN1965, and 'Wildung' was traced to 'Duchess of Oldenburg' (Table S3).

### 3.3.2. LG8 Titratable Acidity QTL

Seven LG8 QTL haplotypes, constructed with fourteen SNP markers spanning a 12.6 cM distance (7.99 Mbp), segregated in the target families for TA and varied significantly in terms of their TA levels ($p < 0.0001$). Three haplotypes (8A, 8B, 8C) were associated with a high relative TA ($\mu = 7.4$ g/L, $\mu = 7.1$ g/L, and $\mu = 7.0$ g/L, respectively), one haplotype (8I) was associated with a low relative TA ($\mu = 5.7$ g/L), and the remaining five haplotypes (8D, 8E, 8F, 8G, 8H) were associated with a moderate TA ($\mu = 6.9$ g/L, $\mu = 6.6$ g/L, $\mu = 6.4$ g/L, $\mu = 6.3$ g/L, and $\mu = 6.4$ g/L, respectively) (Table 2).

The high-TA haplotype 8A of the breeding parent 'MN55' was traced to AA44. The high-TA haplotype 8B of the breeding parents MN1836 and MN1915 was traced to 'Utter's Large Red'. Finally, the high-TA haplotype 8C of the breeding parents 'Honeycrisp' and 'Minneiska' was traced to 'Northern Spy'. The low-TA haplotype 8I of the breeding parents 'Minneiska', 'Minnewashta', and MN1915 was traced to 'Northwest Greening', whereas 8I of MN1965 was traced to 'Montgomery' (Table S3).

### 3.3.3. LG16 Titratable Acidity QTL

Seven LG16 QTL haplotypes, constructed with six SNP markers spanning a 2.0 cM distance (0.38 Mbp), segregated in the target families for TA and varied significantly in terms of their TA levels ($p < 0.0001$). Two haplotypes (16A, 16B) were associated with a high

relative TA ($\mu$ = 6.8 g/L and $\mu$ = 6.8 g/L, respectively), one haplotype (16H) was associated with a low relative TA ($\mu$ = 4.8 g/L), and the five remaining haplotypes (16C, 16D, 16E, 16F, 16G) were associated with a moderate relative TA ($\mu$ = 6.6 g/L, $\mu$ = 6.5 g/L, $\mu$ = 6.0 g/L, $\mu$ = 6.0 g/L, and $\mu$ = 5.9 g/L, respectively) (Table 2).

The high-TA haplotype 16A of the breeding parents 'Minneiska', 'Minnewashta', and MN1836 was traced to 'Duchess of Oldenburg'. The high-TA haplotype 16B of the breeding parents 'Honeycrisp', 'MN55' and MN1965 was also traced to 'Duchess of Oldenburg'. The low-TA haplotype 16H of 'Wildung' was inherited from 'McIntosh' (Table S3).

### 3.4. Haplotype Analyses for Soluble Solids Content QTLs
### 3.4.1. LG1 Soluble Solids Content QTL

The haplotype names and origins follow those assigned for the TA LG1 QTL haplotype analysis. The haplotypes at the LG1 QTL segregated in the target families for SSC and varied significantly in terms of their SSC levels ($p$ < 0.0001). Two haplotypes (1F, 1A) were associated with a high relative SSC ($\mu$ = 14.1 °Brix, $\mu$ = 13.7 °Brix), two haplotypes (1D, 1E) were associated with a low relative SSC ($\mu$ = 13.3 °Brix and $\mu$ = 13.3 °Brix, respectively), and the two remaining haplotypes (1C, 1B) were associated with a moderate relative SSC ($\mu$ = 13.7 °Brix, $\mu$ = 13.5 °Brix, respectively) (Table 3).

**Table 3.** Summary of QTL haplotype effects for the LG1, LG13, and LG16 SSC QTLs. Across-year adjusted SSC means for six families were used.

| LG | Haplotype Name | Haplotype Sequence [z] | Mean (°Brix) | Variance | No. Offspring | Relative Effect |
|---|---|---|---|---|---|---|
| | 1A | ABBABA | 13.7 | 2.3 | 253 | High |
| | 1B | ABBAAA | 13.7 | 1.2 | 39 | Neutral |
| 1 | 1C | BABABB | 13.5 | 1.9 | 153 | Neutral |
| | 1D | BBAABA | 13.3 | 2.3 | 212 | Low |
| | 1E | BBABAB | 13.3 | 2.2 | 152 | Low |
| | 1F | BBABAA | 14.1 | 1.5 | 33 | High |
| | 13A | BABABABABBBAAABBAB | 14.1 | 1.0 | 36 | High |
| | 13B | AAABBAABAAAABBBAAB | 14.1 | 2.3 | 24 | High |
| | 13C | BABABABABAABABAABB | 14.0 | 1.2 | 31 | High |
| | 13D | ABBBABBABBABABAABA | 13.9 | 1.8 | 50 | High |
| 13 | 13E | AAAABBBABBABABAABA | 13.7 | 2.0 | 176 | High |
| | 13F | ABBBBABABAAABBBABA | 13.5 | 1.8 | 25 | High |
| | 13G | AAABBABABAAABBBABA | 13.5 | 2.2 | 263 | High |
| | 13H | AAABBABBAAAAAABBBB | 13.3 | 0.9 | 22 | High |
| | 13I | ABBBABBABAABABAABB | 13.3 | 2.4 | 89 | High |
| | 13J | ABBBABBBAAAAABAABA | 12.3 | 2.0 | 77 | Low |
| | 16A | BABAAB | 13.4 | 2.4 | 201 | Low |
| | 16B | ABBBAA | 13.2 | 2.6 | 204 | Low |
| | 16C | BBBBBB | 13.3 | 2.3 | 145 | Low |
| 16 | 16D | BBAAAB | 13.1 | 2.4 | 136 | Low |
| | 16E | ABBBBA | 14.0 | 2.1 | 45 | High |
| | 16F | BBBBAA | 13.9 | 1.8 | 103 | High |
| | 16G | BBAAAB | 13.9 | 1.2 | 24 | Low |
| | 16H | ABBBAA | 14.1 | 1.3 | 28 | High |

[z] Information on SNP markers used for haplotyping at the LG1, LG13, and LG16 SSC QTLs is presented in Table S2.

The low-SSC haplotype 1D of the breeding parents 'Minnewashta', MN1836, and MN1915 was traced to 'Malinda', whereas for the breeding parent 'MN55' it was traced

to AA44, and for the breeding parent 'Wildung' it was traced to 'Jonathan'. The low-SSC haplotype 1E of the breeding parents 'Honeycrisp', 'Wildung', and MN1915 was traced to 'Duchess of Oldenburg'. The high-SSC haplotype 1A of the breeding parents 'Honeycrisp', 'Minneiska', and 'MN55' was traced to a recombination event between the two haplotypes of 'Grimes Golden'. The high-SSC haplotype 1F of the breeding parent MN1965 was traced to 'Aspa' (Table S3).

### 3.4.2. LG13 Soluble Solids Content QTL

Ten LG13 QTL haplotypes, constructed with eighteen SNPs spanning 12.6 cM (7.99 Mbp) segregated in the target families for SSC and varied significantly in terms of their SSC levels ($p < 0.0001$). One haplotype (13J) was associated with a low relative SSC ($\mu = 12.3$ °Brix) and the nine remaining haplotypes (13A, 13B, 13C, 13D, 13E, 13F, 13G, 13H, 13I) were associated with a high relative SSC ($\mu = 14.1$ °Brix, $\mu = 14.1$ °Brix, $\mu = 14.0$ °Brix, $\mu = 13.9$ °Brix, $\mu = 13.7$ °Brix, $\mu = 13.5$ °Brix, $\mu = 13.5$ °Brix, $\mu = 13.5$ °Brix, and $\mu = 13.5$ °Brix, respectively) (Table 3).

The low-SSC haplotype 13J of the breeding parent 'Minnewashta' was traced to a recombination of the haplotypes of 'State Fair' inherited by 'Minnewashta'. The high-SSC haplotype 13A of the breeding parent 'Wildung' was traced to 'Northwest Greening', the high-SSC haplotype 13B of the breeding parent MN1965 was traced to 'Golden Delicious', the high-SSC haplotype 13C of the breeding parent 'Wildung' was traced to 'Sharon', the high-SSC haplotype 13D of the breeding parent 'MN55' was traced to AA44, the high-SSC haplotype 13E of the breeding parents 'Minneiska', 'Minnewashta', and MN1915 was traced to 'Goodland', the high-SSC haplotype 13F of the breeding parent MN1915 was traced to 'Northern Spy', the high-SSC haplotype 13G of the breeding parents 'Honeycrisp', 'Minneiska', MN1836, MN1965, and 'MN55' was traced to a recombination of the haplotypes of 'Northern Spy' inherited by 'Keepsake', the high-SSC haplotype 13H of the breeding parent MN1836 was traced to a recombination of the haplotypes of 'State Fair' inherited by MN1836, and the high-SSC haplotype 13I of the breeding parent 'Honeycrisp' was traced to 'Duchess of Oldenburg' (Table S3).

### 3.4.3. LG16 Soluble Solids Content QTL

The haplotype names and origins follow those assigned for the TA LG16 QTL haplotype analysis. The haplotypes at the LG16 QTL segregated in the target families for SSC and varied significantly in terms of their SSC levels ($p < 0.0001$). Two haplotypes (16E and 16F) were associated with a high relative SSC ($\mu = 14.0$ °Brix, $\mu = 13.9$ °Brix) and the six remaining haplotypes (16H, 16G, 16A, 16C, 16B, 16D) were associated with a low relative SSC ($\mu = 14.4$ °Brix, $\mu = 13.9$ °Brix, $\mu = 13.4$ °Brix, $\mu = 13.3$ °Brix, $\mu = 13.2$ °Brix, $\mu = 13.1$ °Brix) (Table 3).

The low-SSC haplotype 16C of the breeding parents 'Honeycrisp', MN1836, and MN1915 was traced to 'Frostbite, the low-SSC haplotype 16D of the breeding parents 'Minnewashta', MN1915, and 'Wildung' was traced to 'Northwest Greening', and the low-SSC haplotype 16G of the breeding parent MN1965 was traced to 'Grimes Golden'. The high-SSC haplotype 16E of the breeding parent 'MN55' was traced to AA44. The high-SSC haplotype 16F of breeding parent 'Minneiska' was traced to a recombination of the haplotypes of 'Honeycrisp' inherited by 'Minneiska' (Table S3).

### *3.5. Interactions at and among QTLs*

### 3.5.1. Titratable Acidity QTL Interactions

The interactions among TA QTLs were initially investigated but not reported due to low sample sizes limiting the statistical power. Significant variation among all LG1, LG8, and LG16 functional diplotypes for TA was observed. At all QTLs, the offspring with two copies of a high-TA haplotype had the highest mean, and the offspring with two copies of a low-TA haplotype had the lowest mean. The offspring with at least one neutral-TA haplotype or one high- and one low-TA haplotype varied slightly in their means (Figure 1).

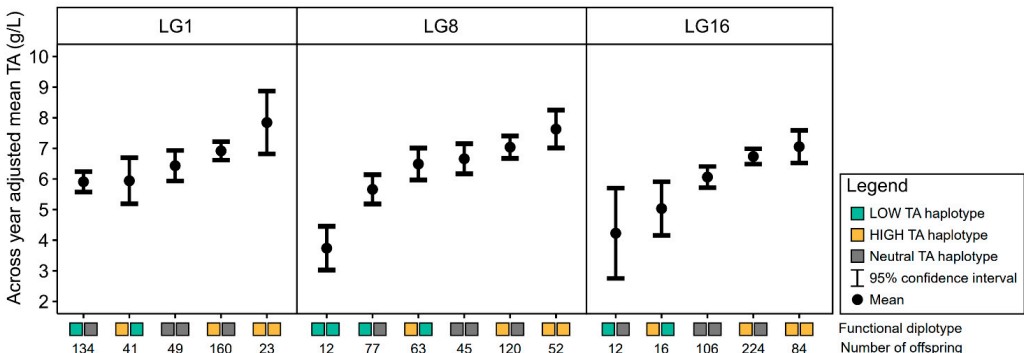

**Figure 1.** Means and 95% confidence intervals for across-year adjusted titratable acidity (TA) for offspring functional diplotypes at the LG1, LG8, and LG16 QTLs. Low-TA haplotypes were 1E, 1F, 8I, and 16H (corresponding to designations in Table 2). High-TA haplotypes were 1A, 8A, 8B, 8C, 16A, and 16B (corresponding to designations in Table 2). The number of offspring with a given functional diplotype is listed below each functional diplotype.

### 3.5.2. Soluble Solids Content QTL Interactions

Similar to TA QTLs, the interactions among SSC QTLs were initially investigated but not reported due to low sample sizes limiting the statistical power. Significant variation among all LG1, LG13, and LG16 functional diplotypes for SSC was observed. At all QTLs, the offspring with two copies of a high-SSC haplotype had the highest mean, and the offspring with two copies of a low-SSC haplotype had the lowest mean. The offspring with at least one neutral-relative-SSC haplotype or one high- and one low-TA haplotype varied slightly in their means, with the exception of the LG13 QTL as there were no offspring with two copies of a low-SSC haplotype (Figure 2).

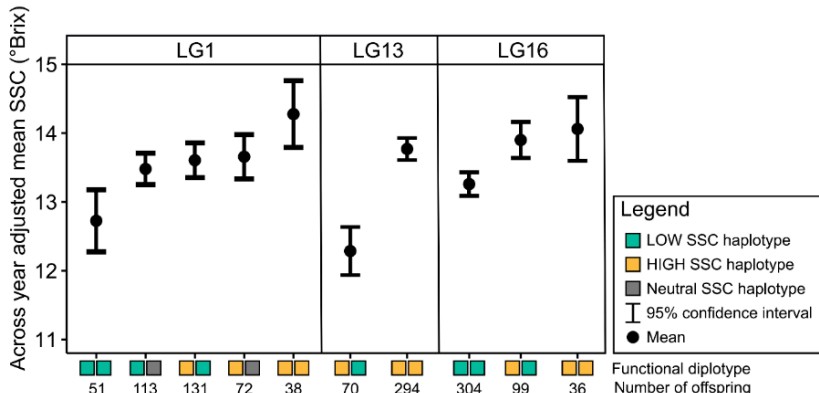

**Figure 2.** Means and 95% confidence intervals for across-year adjusted soluble solids content (SSC) for offspring functional diplotypes at the LG1, LG13, and LG16 QTLs. Low-SSC haplotypes were 1D, 1E, 13J, 16A, 16B, 16C, and 16D (corresponding to designations in Table 3). High-SSC haplotypes were 1A, 1F, 13A, 13B, 13C, 13D, 13E, 13F, 13G, 13H, 13I, 16E, 16F, and 16H (corresponding to designations in Table 3). The number of offspring with a given functional diplotype is listed below each functional diplotype.

## 4. Discussion

Three TA QTLs and three SSC QTLs were characterized across pedigree-connected breeding families representing eight breeding parents. The haplotypes associated with high or low relative trait levels were identified or validated. The genetic information gained in this study will enable more informed selection when targeting TA and/or SSC in apple scion breeding programs that utilize 'Honeycrisp'-derived germplasm.

### 4.1. QTL Identities

The TA QTLs detected in this study colocalized with previously reported QTLs on LGs 1, 8, and 16. The QTL identified on LG1 (*Ma5*, 29.5 Mbp to 30.9 Mbp) has been reported to be associated with TA [9] and sensory acidity [2]. *Ma5* has only been characterized in two half-sib mapping populations [2]. The QTL identified on LG8 (*Ma3*, 8.0 Mbp to 12.8 Mbp) has been well characterized [3,6,7,28–30]. QTL and causal genes have been suggested for the LG8 locus [11]. The QTL identified on LG16 (*Ma*, 0.1 Mbp to 3.4 Mbp) has also been well characterized [3,4,31] and a casual gene responsible for malate transport between the cytosol and vacuole has been proposed [32].

The SSC QTLs in this study colocalized with previously reported QTLs on LGs 1, 13, and 16. The QTL identified on LG1 (29.5 Mbp to 30.9 Mbp) colocalized with the QTLs identified by Guan et al. [5] for fructose (26.6 Mbp to 30.9 Mbp), glucose (26.6 Mbp to 30.9 Mbp), and sorbitol (26.6 Mbp to 30.9 Mbp). The QTL identified on LG13 (15.8 Mbp to 23.8 Mbp) colocalized with the loci identified by Guan et al. [5] for SSC (14.1 Mbp to 23.8 Mbp) and sorbitol (23.4 Mbp to 26.1 Mbp). The QTL identified on LG16 colocalized with the loci identified by Guan et al. [5] for glucose (3.0 Mbp to 5.3 Mbp).

### 4.2. Breeding Parents Heterozygous for the Characterized QTLs

The haplotype analysis findings demonstrated that 'Honeycrisp' was heterozygous for the LG1 TA and SSC QTLs. The 1A and 1E haplotypes of 'Honeycrisp' were traced to 'Golden Delicious' and 'Duchess of Oldenburg', respectively. A breeder would expect offspring from 'Honeycrisp' to segregate for both TA and SSC at the LG1 QTL.

At the LG8 TA QTL, the breeding parents 'Minneiska' and MN1915 had high- and low-TA haplotypes. Both parents had the low-TA haplotype 8I, which was traced to 'Northwest Greening'. 'Minneiska' had the high-TA haplotype 8C, which originated from 'Northern Spy'. MN1915 had the high-TA haplotype 8B, which originated from 'Utter's Large Red'.

The haplotype analysis results indicated that 'Minnewashta', a parent of 'Minneiska', was heterozygous for the SSC LG13 QTL. 'Minnewashta' had the high-SSC haplotype 13E, which originated from 'Goodland' and had the low-SSC haplotype 13J, which originated from 'Minnewashta' as a recombination of the haplotypes of 'State Fair'. 'Minneiska' inherited the high-SSC LG13 haplotype 13E from 'Minnewashta' and therefore, the low-SSC haplotype 13J only segregated in a single family ('Honeycrisp' × 'Minnewashta') in this study.

At the LG16 TA QTL, no breeding parents had both low- and high-effect haplotypes. This is likely due to haplotype 16H, which was inherited by 'Wildung' from 'McIntosh', being classified as the only low-TA haplotype. All other LG16 QTL haplotypes characterized in this study were associated with high or neutral TA levels. In contrast, two breeding parents ('Minneiska', 'MN55') had a high (16E, 16F) and a low (16A, 16B) SSC haplotype at the LG16 QTL. The high-effect haplotype 16E was inherited by 'MN55' from AA44 and high-effect haplotype 16F was inherited by 'Minneiska' from a recombination of the haplotypes of 'Honeycrisp'. The low-effect haplotype 16A was inherited by several parents ('Minneiska', 'Minnewashta', and MN1836) and was traced to 'Duchess of Oldenburg', and the low-effect haplotype 16B that was inherited by several parents ('Honeycrisp', MN1965, and 'MN55') was also traced to 'Duchess of Oldenburg'. These high-effect and low-effect haplotypes appear to have different effects on TA and SSC, which suggests either that the genes affecting the two traits might be in repulsion at the LG16 QTL or pleiotropy.

### 4.3. LG1 QTL Haplotype Putatively Associated with Low Titratable Acidity and High Soluble Solids Content

Most LG1 QTL haplotypes associated with a high TA were also associated with a high SSC; however, haplotype 1F was associated with a low TA and a high SSC. The selection against haplotype 1F is advisable as a high SSC and a low TA often results in insipid fruit. Haplotype 1F was present only in one parent, MN1965, which inherited haplotype 1F from

'Aspa' (Table S3). Therefore, haplotype effects should be considered putative and should be validated in future studies.

### 4.4. LG8 and LG16 Titratable Acidity QTL Haplotype Effects Validated

Haplotypes 8C, 8F, 8H, and 8I at the LG8 QTL and haplotypes 16A, 16B, 16C, 16D, 16E, 16G, and 16H at the LG16 TA QTL were previously characterized by Verma et al. [3] and were validated in this study (Table 2). Verma et al. [3] classified haplotypes at each locus into functional groups where the *q* allele was associated with a low relative TA and the *Q* allele was associated with a high relative TA.

At the LG8 QTL in this study, haplotypes 8F, 8H, and 8I were classified as low-TA haplotypes, which agreed with the functional allele assignments by Verma et al. [3]. Haplotype 8C, which was previously classified as a *q* allele (Table 2) [3], was classified as a high-relative-TA haplotype in this study. This might have been due to the effects of the haplotypes at the LG16 TA QTL. Haplotype 8C was only represented through 'Honeycrisp' in eight small families ($1 < n < 32$) in the Verma et al. [3] paper, whereas in this study it was highly represented in all six families through 'Honeycrisp' or 'Minneiska' ($n = 212$ total individuals, Table S3).

Most LG16 TA QTL haplotypes, except for haplotype 16F, were previously characterized by Verma et al. [3] (Table 2). Most of the haplotype classifications in Verma et al. [3] for LG16 TA QTL haplotypes agreed with the classifications in this study. A limitation to haplotype effect characterization comparisons is that the current study characterized LG16 haplotype effects into three categories (low, neutral, and high) whereas Verma et al. [3] characterized haplotypes into two categories (*Q*/high and *q*/low). The haplotypes that the current study identified as neutral (16C, 16D, 16E, 16F, 16G) were characterized by Verma et al. [3] as "high TA" haplotypes (16C, 16D, 16G) or as a "low TA" haplotype (16E). The remaining haplotype (16F) was not identified by Verma et al. [3] and was characterized by the current study as a neutral-effect haplotype.

### 4.5. Most LG16 QTL Haplotypes Putatively Associated with High Titratable Acidity and Low Soluble Solids Content

Haplotypes at the LG16 QTL (*Ma* locus) associated with a high TA were generally associated with a low SSC. For example, haplotype 16H, which was inherited by 'Wildung' from 'McIntosh', demonstrated the lowest mean TA and the highest mean SSC (Tables 2 and 3). The selection for the high-SSC LG16 haplotypes might result in a low mean TA content and therefore, unbalanced, insipid fruit. In addition, the LG16 QTL colocalized with multiple trait loci at the proximal end of LG16 including bitter pit susceptibility [30], fruit firmness [33], fruit weight [21], and skin color and skin splitting [34]. Due to the importance of LG16 for apple fruit quality traits, breeders should consider the allele effects presented in this study and for all relevant LG16 trait loci when making selection decisions.

### 4.6. Interactions at and among Detected QTLs

The under-representation of some genotype classes hampered the examination of interactions among QTLs. Many of the families were fixed for effects for at least one locus for TA or SSC. Both the TA and SSC means were significantly higher when the offspring had two high-relative-effect haplotypes and were significantly lower when the offspring had two low-relative-effect haplotypes (Figures 1 and 2). The offspring that had at least one neutral-relative-effect haplotype, or one high- and one low-relative-effect haplotype, had moderate trait means compared to the homozygous high or homozygous low groups.

### 4.7. Study Limitations

TA and SSC are strongly influenced by environmental conditions (e.g., temperature) [35] and management practices (e.g., harvest timing) [36]. In this study, most individuals were evaluated in at least two years and year-to-year variation was managed by using across-

year corrected means as phenotypic values. The observed germplasm varied from year to year, contributing to a variation between years in addition to environmental influences.

Small family sizes and an uneven representation of breeding-parent genomes resulted in under-representation or no representation of certain parent haplotypes. The under-representation of some parent haplotypes was not surprising as this is a common challenge in pedigree-based QTL analyses [3,20,25]. Most QTL haplotypes across all TA and SSC QTLs were well represented ($n > 25$), but six haplotypes (8D, 8H, 13B, 13F, 13H, and 16G) were represented by less than 25 offspring. The effects of under-represented haplotypes should be validated in future studies. Additionally, small family sizes made it challenging to examine interactions among QTLs, especially at QTLs with many haplotypes (e.g., LG8, LG13 QTLs).

*4.8. Breeding Implications*

Targeting TA and SSC in breeding programs will continue to be complex as there are no optimal trait levels for either trait, and some QTLs are associated with both traits (e.g., LG1, LG16). For the LG1 QTL, the selection for most high-SSC haplotypes will likely result in the selection for a high TA, except for the 1F haplotype that was associated with a low relative TA and a high relative SSC. Additionally, selecting haplotypes with a high relative TA for the LG16 QTL will generally result in a low relative SSC, except for the 16H haplotype, which demonstrated a low relative TA as well as a high relative SSC.

The LG1 QTL for TA and SSC is of interest for the development of a DNA test due to the breeding relevance of 'Honeycrisp', which was heterozygous for the LG1 QTL. The LG8 QTL and LG16 QTL for TA could be utilized for marker-assisted selection as they were also validated by Verma et al. [3]. Marker tests for these two QTLs have been developed [37], but each program should validate these tests before using them for their germplasm. The use of the LG16 QTL for SSC as a marker-assisted selection test should be further validated for its effects before applying it to a breeding program.

SSC is a quantitative trait likely controlled by multiple small-effect QTLs (e.g., LG1, LG13, LG16). Therefore, genome-wide prediction might be a more appropriate breeding approach when targeting SSC. Previous studies have reported moderate predictive abilities for the genome-wide prediction of SSC in the apple [38,39]. The SSC QTLs detected in this study could be included in genome-wide prediction models as fixed effects. Future studies should estimate the predictive abilities for SSC in 'Honeycrisp'-derived germplasm.

**5. Conclusions**

In this study, three QTLs for TA and three QTLs for SSC were characterized in six families derived from 'Honeycrisp' and 'Minneiska', which are important U.S. apple cultivars and breeding parents. The results from this study will enable more informed parent selection and the development of predictive DNA tests in apple breeding programs that utilize 'Honeycrisp'-derived germplasm. Future studies should validate QTL haplotype effects in other breeding-relevant germplasm and evaluate the potential benefit of utilizing a genome-wide prediction model for SSC.

**Supplementary Materials:** The following supporting information can be downloaded at: https://www.mdpi.com/article/10.3390/agronomy12071703/s1. Table S1: Number of offspring in pedigree-connected full-sib families used in QTL and haplotype analyses. Table S2: SNPs used for haplotype analysis: names (20K array index, NCBI dbSNP accession no, full name, and Affymetrix ID), physical positions (linkage groups and GDDH whole genome sequence coordinates), and genetic positions. Table S3: Haplotype identities assigned via identity with ancestor sources (if known/deduced) and summary information for each haplotype at the LG1, LG8, LG13, and LG16 QTLs. Physical and genetic positions for each SNP are in Table S2.

**Author Contributions:** Conceptualization, J.J.L.; formal analysis, B.A.M.; writing: original draft preparation, B.A.M.; writing: review and editing, S.A.K. and J.J.L. All authors have read and agreed to the published version of the manuscript.

**Funding:** This research was partially funded by State Agricultural Experiment Station—University of Minnesota Projects MIN-21-040 and MIN-21-097 and by the USDA-NIFA Specialty Crop Research Initiative Projects 2009-51181-05808 and 2014-51181-22378.

**Institutional Review Board Statement:** Not applicable.

**Informed Consent Statement:** Not applicable.

**Data Availability Statement:** Not applicable.

**Acknowledgments:** We would like to thank Nicholas Howard for curation of genetic data and assistance in pedigree-based QTL analysis. We would like to acknowledge the many people who assisted with collecting TA and SSC data that provided the phenotyping dataset in this study, including Nicole Marshall, Steven Balaka Opiyo, and Hannah Hauan. We would like to acknowledge the workers at the Horticultural Research Center in Chanhassen, Minnesota, including Winford Mcintosh, for growing and maintaining the trees that were used in this study.

**Conflicts of Interest:** The University of Minnesota receives royalty payments related to the 'Honeycrisp', 'Minnewashta', 'Wildung', 'Minneiska', and 'MN55' apple cultivars. J.J.L., and the University of Minnesota have a royalty interest in these cultivars. These relationships have been reviewed and managed by the University of Minnesota in accordance with its Conflict of Interest policies.

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
