# Peer review of "Large-Effect QTLs for Titratable Acidity and Soluble Solids Content Validated in ‘Honeycrisp’-Derived Apple Germplasm"

_agronomy, doi:10.3390/agronomy12071703_

Round 1

Reviewer 1 Report

MDPI Agronomy

Large Effect QTLs for Titratable Acidity and Soluble Solids 2 Content Validated in ‘Honeycrisp’-Derived Apple Germplasm

Manuscript Number: agronomy-1806428

This manuscript was designed to study a pedigree-based QTL analysis approach to validate QTLs associated with TA and SSC in ‘Honeycrisp’-derived germplasm. The four-year collected data was used leveraged to validate three TA QTLs on linkage groups (LGs) 1, 8, and 16 and three SSC QTLs on LGs 1, 13, and 16. The current results provide valuable insight into potential information to improve quality traits in the apple crop. I feel the content of the study requires further validation. Moreover, the findings are primary results and required further validation.   

I have a few queries that I recommend authors answer and add/improve in the manuscript.

The conclusion part of this manuscript is missing and the authors should add the conclusion part as per the manuscript format of the journal.

The traits used in the study were largely influenced by environmental factors. How these factors were considered for balancing the study

The number of haploids representing the offspring is very less and the study should be extended for proper investigation of a greater number of haploids.

Author Response

1. This manuscript was designed to study a pedigree-based QTL analysis approach to validate QTLs associated with TA and SSC in ‘Honeycrisp’-derived germplasm. The four-year collected data was used leveraged to validate three TA QTLs on linkage groups (LGs) 1, 8, and 16 and three SSC QTLs on LGs 1, 13, and 16. The current results provide valuable insight into potential information to improve quality traits in the apple crop. I feel the content of the study requires further validation. Moreover, the findings are primary results and required further validation. I have a few queries that I recommend authors answer and add/improve in the manuscript.

Response. Thank you for reviewing the manuscript and providing feedback. We appreciate your time and thoughtful review.

2. The conclusion part of this manuscript is missing and the authors should add the conclusion part as per the manuscript format of the journal.

Response. A conclusion section has been added to the manuscript.

Updated text (lines 421-428)

In this study, three QTLs for TA and three QTLs for SSC were characterized in six families derived from ‘Honeycrisp’ and ‘Minneiska’, which are important U.S. apple cultivars and breeding parents. The results from this study will enable more informed parent selection and development of predictive DNA tests in apple breeding programs that utilize ‘Honeycrisp’-derived germplasm. Future studies should validate QTL haplotype effects in other breeding-relevant germplasm and evaluate the potential benefit of utilizing a genome-wide prediction model for SSC.

3. The traits used in the study were largely influenced by environmental factors. How these factors were considered for balancing the study

Response. Thank you for this comment. Titratable acidity and soluble solids content are influenced by environmental conditions as described in the discussion (lines 386 to 390). Most individuals in this study were harvested in multiple years, which enabled estimation of year effects. We accounted for year-to-year variation by utilizing corrected means in QTL analyses as described in the methods (lines 82 to 84).

4. The number of haploids representing the offspring is very less and the study should be extended for proper investigation of a greater number of haploids.

Response. Thank you for this comment. The germplasm for this study was selected because of its breeding relevance to the University of Minnesota apple breeding program and many other breeding programs we are aware of that are using ‘Honeycrisp’-derived germplasm. Therefore, we focused on estimating effects of haplotypes derived from ‘Honeycrisp’ and ‘Minneiska’. As described in the study limitations (lines 394 to 396), most QTL haplotypes across all TA and SSC QTLs were well represented with at least 25 offspring. There were six haplotypes that were represented by less than 25 offspring. We state in our limitations that haplotypes represented by less than 25 offspring should be targeted in future studies (lines 394 to 396).

Reviewer 2 Report

The authors describe QTL analysis of titratable acidity and soluble solids in apple germplasms. The ms was basically well written. Multiple QTLs were identified, but I felt that the reproductivity of the detected QTLs were unclear. Overall, I would think that the ms could be acceptable for publication in agronomy after corrections according to the following comments.

Major comment
The current QTL analysis was conducted with FlexQTL. How are such the QTLs also detectable in the family (those of >70 offsprings) separately, with general methods for QTL analysis (pseudo-test cross with MapQTL, QTL Cartographer, MapMaker, etc.)? Especially, are the QTLs reproductive with the replicated phenotypic data in the family Honeycrisp x Minnewashta?

Minor comment
L72: Degrees should be indicated in centigrade instead of Fahrenheit.

Author Response

1. The authors describe QTL analysis of titratable acidity and soluble solids in apple germplasms. The ms was basically well written. Multiple QTLs were identified, but I felt that the reproductivity of the detected QTLs were unclear. Overall, I would think that the ms could be acceptable for publication in agronomy after corrections according to the following comments.

Response. Thank you for taking the time to review this manuscript. We appreciate your feedback and comments.

2. Major comment: The current QTL analysis was conducted with FlexQTL. How are such the QTLs also detectable in the family (those of >70 offsprings) separately, with general methods for QTL analysis (pseudo-test cross with MapQTL, QTL Cartographer, MapMaker, etc.)? Especially, are the QTLs reproductive with the replicated phenotypic data in the family Honeycrisp x Minnewashta?

Response. Thank you for this suggestion. Most QTLs reported in this study colocalize with titratable acidity and soluble solids content QTLs previously reported in the literature (see Discussion lines 289 to 303) which indicates that these QTLs are repeatable in other germplasm sets and environments.  Additionally, these QTLs were detected in multiple years and in the corrected means datasets further demonstrating their repeatability. In this study, a pedigree-based QTL analysis (PBA) approach was used due to the highly interconnected nature of the breeding germplasm. The use of pedigree-connected germplasm increases statistical power for QTL detection and has been commonly used for various traits in Rosaceae crops such as apple, peach, and sweet cherry (see example citations below). FlexQTLTM software was designed to carryout PBA in complex germplasm sets like the one described in this study.

Apple examples:

  • Allard, A. et al. Detecting QTLs and putative candidate genes involved in budbreak and flowering time in an apple multiparental population.  Exp. Bot.67, 2875–2888 (2016). https://doi.org/10.1093/jxb/erw130
  • Bink, M. C. A. M. et al. Bayesian QTL analyses using pedigreed families of an outcrossing species, with application to fruit firmness in apple.  Appl. Genet.127, 1073–1090 (2014). https://doi.org/10.1007/s00122-014-2281-3
  • Di Guardo, M. et al. Deciphering the genetic control of fruit texture in apple by multiple family-based analysis and genome-wide association.  Exp. Bot.68, 1451–1466 (2017). https://doi.org/10.1093/jxb/erx017
  • Guan, Y., Peace, C., Rudell, D., Verma, S. & Evans, K. QTLs detected for individual sugars and soluble solids content in apple.  Breed.35, 334 (2015). https://doi.org/10.1007/s11032-015-0334-1
  • Howard, N. P. et al. Two QTL characterized for soft scald and soggy breakdown in apple (Malus× domestica) through pedigree-based analysis of a large population of interconnected families. Tree Genet. Genomes 14, 2 (2018). https://doi.org/10.1007/s11295-017-1216-y
  • Kostick, S.A., Teh, S.L., Norelli, J.L. et al.Fire blight QTL analysis in a multi-family apple population identifies a reduced-susceptibility allele in ‘Honeycrisp’. Hortic Res 8, 28 (2021). https://doi.org/10.1038/s41438-021-00466-6
  • van de Weg et al. Epistatic fire blight resistance QTL alleles in the apple cultivar ‘Enterprise’ and selection X-6398 dsicovered and characterized through pedigree-informed analysis. Mol. Breed. 38, 5 (2018). https://doi.org/10.1007/s11032-017-0755-0
  • Verma, S. et al. Two large-effect QTLs, Maand Ma3, determine genetic potential for acidity in apple fruit: breeding insights from a multi-family study. Tree Genet. Genomes 15, 18 (2019). https://doi.org/10.1007/s11295-019-1324-y

Peach examples:

  • Fresnedo-Ramírez, J., Bink, M.C.A.M., van de Weg, E. et al. QTL mapping of pomological traits in peach and related species breeding germplasm. Mol Breeding35, 166 (2015). https://doi.org/10.1007/s11032-015-0357-7
  • Rawandoozi, Z.J., Hartmann, T.P., Carpenedo, S. et al. Identification and characterization of QTLs for fruit quality traits in peach through a multi-family approach. BMC Genomics 21, 522 (2020). https://doi.org/10.1186/s12864-020-06927-x
  • Rawandoozi, Z.J., Hartmann, T.P., Carpenedo, S. et al. Mapping and characterization QTLs for phenological traits in seven pedigree-connected peach families. BMC Genomics 22, 187 (2021). https://doi.org/10.1186/s12864-021-07483-8

Sweet cherry examples:

  • Cai, L., Quero-García, J., Barreneche, T. et al. A fruit firmness QTL identified on linkage group 4 in sweet cherry (Prunus avium L.) is associated with domesticated and bred germplasm. Sci Rep 9, 5008 (2019). https://doi.org/10.1038/s41598-019-41484-8
  • Crump, W.W., Peace, C., Zhang, Z. et al. Detection of breeding-relevant fruit cracking and fruit firmness quantitative trait loci in sweet cherry via pedigree-based and genome-wide association approaches. https://doi.org/3389/fpls.2022.823250

3. Minor comment: L72: Degrees should be indicated in centigrade instead of Fahrenheit.

Response. The authors have made the suggested edit the text to change the degrees to indicate centigrade instead of Fahrenheit.

Updated text (line 72)

Original text: “Juice samples were thawed to room temperature (70°F) and homogenized prior to…

Updated text: “Juice samples were thawed to room temperature (21°C) and homogenized prior to…”.